# Neural Representation of Costs and Rewards in Decision Making

**DOI:** 10.3390/brainsci11081096

**Published:** 2021-08-20

**Authors:** Yixuan Chen

**Affiliations:** Queens’ College, University of Cambridge, Cambridgeshire CB3 9ET, UK; yc471@cam.ac.uk

**Keywords:** decision making, dopamine, cost and reward encoding

## Abstract

Decision making is crucial for animal survival because the choices they make based on their current situation could influence their future rewards and could have potential costs. This review summarises recent developments in decision making, discusses how rewards and costs could be encoded in the brain, and how different options are compared such that the most optimal one is chosen. The reward and cost are mainly encoded by the forebrain structures (e.g., anterior cingulate cortex, orbitofrontal cortex), and their value is updated through learning. The recent development on dopamine and the lateral habenula’s role in reporting prediction errors and instructing learning will be emphasised. The importance of dopamine in powering the choice and accounting for the internal state will also be discussed. While the orbitofrontal cortex is the place where the state values are stored, the anterior cingulate cortex is more important when the environment is volatile. All of these structures compare different attributes of the task simultaneously, and the local competition of different neuronal networks allows for the selection of the most appropriate one. Therefore, the total value of the task is not encoded as a scalar quantity in the brain but, instead, as an emergent phenomenon, arising from the computation at different brain regions.

## 1. Introduction

The elaborate neural computation in our brain that is used for decision making is a process that people take for granted every day. We decide what to eat, whether to go shopping, and whether to spend money on a lottery ticket without realising the massively complicated computation that takes place in our brains. To make a good decision, an agent has to learn the cost and benefit of each attribute of their choice. Moreover, the internal states (e.g., whether an animal is hungry) and the agent’s goal should also be considered. When the decision has been made, the task may require the agent to select the appropriate actions, bringing action selection into the picture. Combining all of these factors, making a decision alone seems like a rather miraculous thing for brains to do, and even more miraculously, all of these computations are conducted rapidly. Although the choice might not always render the optimal results and even though people do make bad choices, this only makes the way that the brain makes decisions more fascinating.

There are many different types of decisions that people make in everyday life. For example, when the environment is constant, a choice is made with all of the information presented simultaneously, and this is called a within-state decision. When the environment and states change with time, this means that the best option a minute ago might no longer be optimal now. Decisions made across time with changing environments are called state-change decisions. Decisions can also be model-free or model-based. A model-free decision, as its name suggests, does not require the modelling of the external world and is heavily dependent on feedback. A choice could be made through simple reinforcement learning, which follows the “law of effect”: those actions that are rewarded will be more likely to be repeated in the future [1]. A model-based decision, on the other hand, relies on the modelling of external events and the computation of the probabilities of different alternatives such that a choice could be made by comparing different attributes in different options [2,3]. This means that an animal’s decision may not be rigidly habitual and that goal-directed behaviours with a greater level of flexibility are possible in a volatile environment. For both types of decisions, either model-free or model-based, the value of each attribute or action is updated with experience, and learning happens concurrently [4]. These different choices recruit different brain regions under the appropriate circumstances, and they will be discussed in detail later in this article.

This review begins with how reward is encoded in the brain. While the midbrain dopaminergic system encodes reward prediction errors and motivational cues [5,6], the actual prediction of a reward is more related to the forebrain structures [7,8,9], and its value is updated with learning, possibly through the error signalled by the dopaminergic system. There are a few suggestions about how the two signals encoded by dopaminergic neurons are differentiated by the downstream structure [10,11]; however, it is under debate. How costs are encoded in the brain is then discussed, and a system that is the ‘opposite’ of the dopaminergic neurons for punishment prediction error in the lateral habenula is introduced [12,13]. An overlapping pattern of reward and cost encoding suggests that the forebrain structures are also important in choice selection. When it comes to deciding what the ultimate choice might be, these brain regions do not work in isolation. A decision is made through an interconnected, distributed, recurrent network with an emergent property [14,15], and choices are weighted by internal states at the nucleus accumbens before action selection [16].

## 2. Reward Encoding and Learning

Before making any decision, an animal should be able to encode the rewards and costs in the brain such that they are able to make good choices that give them a higher chance of survival. Therefore, the way that this information has been stored in the brain and the retrieval of relevant memories are crucially important. In the last few decades, research on the ‘reward system’ has made quite a lot of progress, and the importance of dopaminergic system has been greatly emphasised [17]. However, it is now known that the involvement of the dopaminergic system in the reward encoding process does not mean that this system gives animals pleasurable feelings [18,19,20]. Later research has suggested that, apart from dopaminergic system, other brain areas such as the anterior cingulate cortex (ACC), prefrontal cortex, and limbic systems, etc., are also involved in the encoding of rewards for the decision-making process [21,22,23,24]. Finding multiple brain regions that could encode the reward does not imply the failure of seeking a congruent pattern with incongruent results. Instead, it demonstrates the complexity of the decision-making process, which requires information from different aspects, including the internal bodily state, external information, and past experiences.

### 2.1. Dopamine and Model-Free Choices

Model-free reinforcement learning updates the value of an action or stimulus based on feedback (essentially prediction error) and is used to directly guide behaviours. A model-free choice creates responses more quickly, but it is also more rigid. The dopaminergic system is very important in model-free decision making in that it encodes the motivational cue and reward prediction error for learning [5,6], but other brain regions, such as the amygdala, are central to model-free learning as well [25]. This section discusses the importance of dopamine in powering model-free choices and instructing learning when a prediction error has occurred.

Reward processes can be divided into three major components: liking (subjective experience and expression of pleasure), wanting (motivational component for overcoming the costs to obtain the rewards), and learning (e.g., Pavlovian and instrumental conditioning) [18]. It has been widely, but most likely erroneously, believed by the public that dopamine level in the brain is essential for the ‘liking’ component of a reward. The ‘dopamine anhedonia hypothesis’ popularised in 1970s states that anhedonia is correlated with a low level of dopamine in the brain [26]. This view has subsequently been challenged, and it is now generally believed by the field that dopamine is more responsible for the motivational and learning component of rewards (i.e., wanting and learning) and that it is less involved in subjective feelings and expressions of enjoyment [18].

Early evidence mainly comes from selective lesion studies of dopaminergic neurons in model organisms and human patients. The majority of dopaminergic neurons are located in mesolimbic dopamine system, which is prominent for incentive motivation [27], and the nigrostriatal dopamine system, which converts motivation into actions [28]. Selective damage to the dopaminergic system by the microinfusion of oxidopamine (6-OHDA), which removes 99% of the dopamine in the nucleus accumbens and neostriatum (caudate and putamen), caused rats to have severe motivational and motor response deficits, but the consummatory responses, such as the ability to swallow the sucrose solution provided to them, and the subsequent ‘liking’ reaction were intact [19]. 

Similarly, human studies have also shown deficits in motivation but less deficits in hedonic feelings towards sweet tastes in Parkinson’s patients who have extensive dopaminergic neuron degeneration [29]. While their hedonic responses to primary reinforcers are largely intact, some Parkinson’s patients show deficits in motivation (e.g., apathy), which could be improved by dopamine replacement therapy [20]. However, this therapy may sometimes lead to impulse control problems in patients, including gambling, disinhibited sexual behaviours, compulsive eating, etc. However, these patients do not report high levels of pleasure while engaging in these activities [30,31,32]. In other words, lesions in dopamine system cause animals and/or people to be less motivated to procure the rewards and the primary reinforcers even though the actual rewards still make them feel good (‘like but do not want’). Therefore, the dopaminergic system is more related to the representation of reward-related stimuli and motivation, but it is less about the experience of pleasurable feelings [18,19]. In the human brain, the regions that have been shown to be closely correlated with subjective pleasure rating include the orbitofrontal cortex (OFC), the mid-anterior OFC in particular, and the nucleus accumbens (NAc) [18]. However, perhaps surprisingly, the proportion of NAc that enhances the liking response is only 10% of its total volume, and these neurons are largely GABAergic neurons that are located rostrally [33]. This implies that NAc has other important roles.

The exclusion of dopaminergic neurons from the ‘liking’ response does not necessarily exclude them from the decision-making process. The role of dopamine in reward encoding is less straightforward, as it could be difficult to dissociate motivation from a subjective perception of value. It used to be believed that midbrain dopaminergic neurons (ventral segmental area, VTA) encode reward quantitatively in their phasic firing rate [34,35]. Interestingly, these neurons also encode the probability of acquiring the future reward [14], but they still fail to differentiate the motivational cue from subjective value perception. Fiorillo and his team [36] have shown that the existence of a dopamine ramp in the VTA dopaminergic neurons during the time between the perception of conditioned stimuli and the delivery of an actual reward when the stimuli predicted that the probability of the future reward was between 0 and 1.0. Although the results seem to suggest dopaminergic neurons in the VTA of the mesolimbic dopamine system represent the ‘value’ of rewards and reward-related stimuli, they could also be motivational cues to direct behaviours: if the reward is highly valued then it is probably worth spending time and energy to obtain.

In fact, later studies have shown that the dopamine is more important in motivation to energise actions than simply representing the reward value. The mesolimbic dopamine was released when movements were required to procure rewards, but dopamine did not increase if stillness was required to obtain the rewards [5]. Moreover, the firing of VTA neurons is maintained until the correct actions are executed, indicating its importance in maintaining motivation and powering the movement execution [5]. This effect is very important in drug abuse, as addictive drugs act on the mesolimbic dopaminergic system and cause the overvaluation of drug rewards [37]. In the presence of drug-related stimuli, patients feel the compulsion to obtain the drug while experiencing an elevated dopamine level, even though the drug may not necessarily be as rewarding [4,16].

However, the motivational value represented by dopamine has to be updated with experience, which draws the importance of learning. Learning should occur when the results are different from the expectation. The experiment conducted by Schultz et al. in 1997 demonstrated that the burst of the firing pattern in the midbrain dopaminergic neurons was only seen in the presence of unexpected rewards or in the presence of the conditioned stimuli that predict the future presence of the rewards, but not if they had been expected [6]. Schultz et al. concluded that the phasic firing of the dopaminergic neurons represents the prediction error of whether the reward will appear in the future and not simply the presence of reward. The prediction error signals the imperfection in the predicting system and more learning is required. Up until recently, the reward prediction error has always been believed to be represented as a single scalar value. Dabney et al. suggested the distributional coding of the reward prediction error in the VTA, with some dopaminergic neurons being more ‘optimistic’ and responding positively to almost all rewards, while some neurons were more ‘pessimistic’ and only responding positively only if the reward was very large [38]. This model fits well with actual neural recording in mice VTA [38]. It thus appears that the prediction error signal is not encoded as an average in every dopaminergic neuron but instead as a probability distribution across a population.

This finding, although is significant, posits an encoding problem by the dopaminergic system: does the phasic firing encode motivation to obtain the rewards or the prediction error which leads to learning? To answer this question, some have proposed that a tonic change in dopamine level represents motivation, while the phasic rapid dopamine release encodes prediction error [10]. This has been a matter of debate for some time [39,40] and recently, some progress has been made. Berke proposed that the motivation is encoded by the rapid phasic firing of the midbrain dopaminergic neurons but that learning occurs with the synaptic modulation of the baseline dopamine level by reward prediction error [11]. Those stimuli with a lower reward expectation and lower motivational value have lower baseline dopamine and vice versa. To avoid confusion between the prediction error and motivational signals, which are both encoded by rapid phasic firing of dopaminergic neurons, striatal cholinergic interneurons may switch the dopaminergic synapses to the ‘learning mode’ to induce synaptic plasticity if the stimuli are unexpected but not if the neurons encode motivational cues. These neurons are possibly activated by the VTA GABAergic neurons [41] or by the ‘surprise’-related cells in the intralaminar thalamus [42]. This allows dopamine to encode the overall reward value and the motivational signals that are subject to change based on experience.

However, some recent research has claimed that the dopamine ramp may not necessarily be the motivational signal after all, as it could be explained by the prediction error [43,44,45]. When mice were trained to navigate through a maze in order to attain the reward, a gradual dopamine ramp, instead of phasic firing pattern, was seen in the VTA as the mice gradually approached the reward [46]. This finding was previously used as a supporting evidence for mesolimbic dopamine as motivational cues, and the mice were simply becoming more motivated to proceed as they approached the reward, i.e., when the chance of actually receiving the reward became higher. Kim et al. argued the contrary: the mice being teleported closer to rewards showed a change in the dopaminergic neuron firing pattern, which was better modelled by the prediction error in the temporal difference model, instead of the value [45]. To further support this idea of the prediction idea, the ramp in VTA neuron firing has also been found across studies, indicating that the dopamine ramp at the nucleus accumbens (NAc) is caused by VTA neuron firing [47].

According to Berke’s argument [11], the dopamine ramp itself should not be perceived as a ‘tonic’ elevation of dopamine concentration, as it only lasts for several seconds. Instead, they should be regarded as a continuous increase in motivation. This directly contradicts the findings of Kim et al. even though they ignored the baseline dopamine level at the synapses and only focused on the relative change of the dopamine level in the ventral striatum. This is because Kim et al. also found that the rate of change of the reward prediction error could better model the gradient of dopaminergic neuron firing, which was unaffected by the baseline dopamine [45]. Thus, how the downstream structure of these neurons could differentiate these signals is still a matter of debate. 

The VTA dopaminergic neurons project to many other brain areas apart from the nucleus accumbens (NAc), including the ACC, OFC, hippocampus, and amygdala, and these projections come from different subsets of dopaminergic neurons [48]. The reward prediction error encoded by the VTA neurons is preserved in the dopamine released in NAc, but dopamine release is less associated with cost encoding [49]. These dopaminergic neurons either inhibit NAc medium spiny GABAergic neurons (MSN) by binding to the D1-receptors or activate NAc by binding to the D2-receptors on MSNs [50]. As discussed later in this article, NAc seems to be a place where appetitive and aversive stimuli converge and is an important place for decision making. Their neurons do not simply encode the reward from one stimulus and are more similar to a place where the internal states and perceived values are integrated [16].

### 2.2. Other Brain Areas and Model-Based Choices

So far, the experiments we have discussed are largely based on model-free reinforcement learning, including classical Pavlovian conditioning and instrumental conditioning. Nevertheless, in everyday life, people also use model-based decision making by simulating the external world such that a choice could be made through the comparison of different alternatives. This method, though more computationally difficult, is more flexible and allows for long-term outcomes to be considered. Model-based reinforcement learning stores and updates different aspects of actions and states. Goal-directed behaviour is largely led by model-based reinforcement learning in that the consequences of different actions need to be compared so that the optimal one can be selected and executed. Thus, it is not surprising that other brain areas are involved in this type of decision making, as explicit memories and, most likely conscious, interpretation of states and actions are required. Across neuroimaging, lesion, and electrophysiological studies, the cingulate and prefrontal cortex (including orbitofrontal cortex, ventromedial prefrontal cortex, prelimbic areas, etc.) seem to be very important in model-based decision making [7,8,9]. However, recent studies suggest that dopamine cannot be totally excluded from the picture, and the prediction error may also serve to update the value for actions and states [51,52,53]. These brain areas are shown in Figure 1 below [54].

Before discussing how other brain areas are involved in reward encoding, it should be noted that this knowledge is not as well-established as the information regarding the dopamine system. This is partially due to the fact that the studies on brain areas such as the prefrontal cortex rely heavily on primates since these regions are not as developed in rodents, and the use of PFC anatomical terms in rodents has little consensus [55]. In human, the methods that are recruited are mainly neuroimaging and (pathological) lesion studies. This could potentially be problematic, as the findings are primarily correlation and not causation.

The orbitofrontal cortex (OFC) and anterior cingulate cortex (ACC) have long been shown to have activities that are correlated with task value and that are closely involved in decision making [OFC: 21,22; ACC: 23,24]. However, the differences between these two regions are revealed by their macrocircuit: OFC shows strong connections to areas in the ventral visual stream and the temporal lobe, which contain highly processed sensory information; ACC, however, shows strong connections to the dorsal striatum (putamen and caudate) and supplementary motor areas [14]. This means that the OFC possibly converges the multimodal sensory information for reward prediction, while the ACC may guide action selection to obtain rewards. 

The neurons in the OFC encode future rewards in an identity-specific fashion [56], which is important in model-based decision making. The OFC neurons do not simply represent value in a unitary fashion; it can also represent different aspects of the expected rewards, including the taste, smell, visual appearance, and texture of the reward [57,58]. Moreover, identity specificity may change over different contexts, and new tasks may be encoded by the same neurons [21]. As for the ACC, although there is a lot of evidence showing its significance in effort-based decision making [59,60], a different set of neurons in this area do appear to encode the reward value and respond to reward delivery [24]. For those neurons with activity that is correlated with the predicted task value, a pattern can be seen as to whether the value is modulated by probability or reward size [24]. Interestingly, the connections between the VTA and ACC appear to be important outcome learning and action execution, with the ACC-to-VTA signalling being involved in anticipatory choice and the VTA-to-ACC signalling being involved in error detection [23]. This is consistent with the idea that ACC is a site for action selection for the acquisition of rewards.

The role that vmPFC plays in reward encoding seems to be more complicated, and this is partly due to the fact that it is a region containing three distinct divisions: the medial OFC, the ventral cingulate cortex, and the posterior frontopolar cortex [61]. The vmPFC encodes reward value regardless of whether the physical motor response is required or not, indicating that its role is more involved in value representation and is less involved in action selection or motivation [62]. However, it is more commonly accepted that the vmPFC is a region where combined value is calculated [63,64] and human emotional states are affected [65]. Moreover, regional differentiation has been found in monkey vmPFC, with the ventral neurons preferring appetitive-related stimuli and their value, while the dorsal neurons are predominantly aversive-preferring [66]. Thus, it could be that the combined value is computed from these neurons. This role of the vmPFC will be discussed in detail later in this article.

### 2.3. Updating the Reward Value with Learning

Perhaps surprisingly, dopamine may also play a role in model-based learning. The value of each action and state needs to be updated through learning, and dopamine seems to contribute to the learning of these values [65]. Apart from encoding reward prediction error, some researchers have found that dopamine seems to encode sensory prediction error which represents surprising events that are not necessarily better or worse than expected [67,68]. It thus may have the potential to broadcast a multidimensional model-based prediction error signal that teaches not only the value but also the state, identity, and timing aspect of reward prediction [53]. In fact, dopamine seems to be able to encode an array of sensory, motor, and cognitive variables in mice, including the movement speed and acceleration, object position, view angle, etc. [69]. However, this idea has been challenged by Gorhn et al. [70], who stated that the neutral surprise signal was not found in the striatum or VTA but in lateral PFC. When monkeys learned to respond to a visual stimulus, one or three drops juice were delivered with equally high probability. This made the monkeys expect an average of two drops of juice, but the probability of actually receiving two drops of juice remained very low. Thus, the experimenters constructed three surprising events: the classical reward surprise, when more or less juice than expected was given; the rare reward surprise, when two drops of juice were given; and the visuospatial surprise, when the stimulus appeared at a different location. The fMRI results showed that the ventral striatum and VTA were only involved in classical reward amount surprise trial but not in the other trials. Instead, the lPFC appeared to be active for all types of surprise events that deviate from expectations, which might be expected from a domain-general learning mechanism. On the other hand, the OFC was only active in rare reward surprise trial, when the environment was more learnable than it was in the visuospatial surprise trial. The debate of whether dopamine could encode the surprise-related prediction error is ongoing.

Even if dopamine does not broadcast the surprise signal for model-based learning, it could still at least facilitate the update of information in other brain regions with the reward prediction error. This is useful because dopaminergic projections to higher-order brain regions such as the prefrontal cortex (PFC) can influence attentional control [71], working memory [72], and learning [73,74] following reward-associated surprise events such that future predictions could be more accurate. More direct evidence of dopamine involvement in model-based learning comes from neuroimaging studies of BOLD signals using fMRI [52], which showed that the ventral striatum and vmPFC are both activated in model-free and model-based tasks.

## 3. Cost Encoding and Learning

Before pursuing a reward, one has to determine the cost or the effort that must be exerted in order to attain the reward. Even a task as simple as waiting for the delivery of a reward imposes potential costs to animals since time is of great value. For example, the longer the time spent on foraging, the more likely that animals will be caught by predators in the wild. The cost only becomes negligible when the reward is delivered immediately and directly to the animals for consumption. Therefore, the cost must be represented or encoded by the brain in a way that the most appropriate option can be chosen. The way that neurons encode costs and rewards is tricky to dissociate because the costs could be encoded as a reduction in the activity of the neurons encoding the reward or as an increase or a reduction in the activity of a separate group of neurons encoding the costs. Moreover, the cost, similar to many other attributes of a choice, might not be encoded by single neurons but through population coding [38], which makes finding its neural representation even more difficult.

Moreover, the effort paradox argues that the definition of effort as the activities that are costly contradicts many animal behaviours, as sometimes effort is also valued [75]. For example, contra-freeloading describes the situation in which animals choose to work for food even when the food is directly available to them [76]. In humans, effort being valued is not uncommon as well, and people appear to value both the effort and the product of their effort [75]. This counter-intuitive phenomenon further complicates the study of costs, but in this article, the word ‘cost’ is limited to the energy expenditure (i.e., effort) to procure the reward and the drop in the possibility of receiving future rewards that are viewed as costly but not valued. Although the research is not as rich as that for the study of reward, it seems that most of the brain regions that are responsive to the anticipation of rewards also respond to cost anticipation [59,65,77,78]. In this way, the cost and benefit of each action does not need to converge to a new area, and there will be less redundancy in value computation.

The OFC neurons discussed above encode the identity and value of rewards [62], but they do not just simply encode the reward value for any given sensory stimuli. The first evidence probably comes from lesion studies: rats with OFC lesions appear to be prone to acquiring immediate rewards instead of higher rewards with a longer delay [22,79]. In other words, the OFC-lesioned rats are more cost-aversive. However, lesion studies can be confounded by many of variables (e.g., the lesion site might not actually be the place where the experimenters assumed it to be; lesion could destroy the axons passing through that area; diaschisis). Electrophysiology study provides more direct evidence of the OFC neurons encoding the temporal costs of a choice as a reduction in phasic firing [57]. A cellular recording of rat OFC neurons showed that neurons are more active when the coming rewards would be delivered after a short delay versus those that would arrive after a longer delay, irrespective of the absolute reward size [57]. In this study, the cells also retain the response selectivity towards reward stimuli. When taken together, it seems that the temporal cost and reward value are encoded by different groups of neurons, allowing the temporal cost to be compared to the actual value of reward in the OFC. It could be that OFC lesions remove such a comparison, and the immediate reward becomes more desirable than the otherwise more valuable long-delay choice.

However, the behaviour of lesioned rats could also be explained by the fact that OFC represents the current state of the task that may not be evident from immediate sensory information [65]. The lesions in the OFC prevented the rats from accessing the inferred state representation of the long-delay alternative, which had a higher reward outcome that was not apparent from the immediate environment, while the ability to access the state representation based on immediate observable information through striatum was still intact. Therefore, the rats preferred lower the cost option with a shorter delay. This explanation is compatible with the current literature on the OFC as a structure found to encode the functions related to decision making: expected choice outcome, choice, recent history of choice [80], response inhibition, somatic marker [81,82], etc. The downstream circuits then readout the particular task variable through the stable population representation in the OFC [80]. This is in line with the idea that OFC is a cognitive map, a multivariate store representing the cognitive and behavioural task states relevant to the current goals [83,84]. The stored information can be accessed even when the information is not obvious from immediate external stimuli.

The importance of ACC in effort-based decision making has also been emphasised across many studies, as effort-demanding tasks, either cognitively or physically, seem to correlate with higher ACC activation [59,60]. As ACC also corresponds to the reward signals, we cannot yet conclude that these neurons only encode the effort of a task. Compared to OFC lesion studies, ACC lesions show similar but distinct patterns of responses [85]: rats with ACC lesions appeared to be effort-aversive, as they would favour lower-effort/low-reward choices over high-effort/high-reward alternatives, but their delay-based decision-making ability was unaffected [86,87]. There seems to be a double dissociation between the effort and the temporal delay encoded by the ACC and the OFC, respectively. More interestingly, a recent finding has shown that the ACC neurons were essential in allocating cost to different options with a given budget, and the lesions in the ACC caused the animals to fail to maximise their choice utility by considering the budget available to them [88]. This finding implies that ACC is not only important in encoding the costs; the information it stores is also essential in assigning the effort and cost of a decision. Nevertheless, as mentioned above, the information given by the lesion study, although very valuable, is still limited. 

Hillman and Bilkey [89] investigated how competitive effort (the presence of a conspecific which competes for the resources) affected the firing rate of ACC neurons. If the neurons encoded the effort itself, then they would be more active when competition existed. Instead, Hillman and Bilkey found that the neurons were the most active when the choice had maximum utility. This idea is also supported by neuroimaging, which suggests that the activation of the ACC was elicited by an upcoming difficult task that had higher reward prospects [59]. Therefore, the ACC appears to be a place where the costs and rewards are compared instead of a place where only the cost or reward is encoded. The neural connection between the OFC and the ACC further supports this idea and the OFC may convey important information based on sensory input for decision making in the ACC: the local field potential reveals neural synchronisation in the theta/low beta frequency bands between the OFC and ACC when animals prefer high-effort/high-reward options [90]. 

Going beyond the prefrontal and cingulate cortex, the NAc also seems to participate in effort-related processes [77,78]. With the dopaminergic input from the VTA and glutamate input from regions such as the PFC, amygdala, and hippocampus onto the MSNs, cost could be encoded by activating these GABAergic MSNs to enhance their inhibition of other brain areas [77]. For example, one of the brain areas that MSN releases GABA to is the ventral palladium (VP), which shows a very low firing rate after the administration of hypertonic saline, an aversive stimulus [91]. Both NAc and VP are downstream components of decision making, so it is likely that the cost they encode is value preserved from higher brain regions that are subject to modification. Moreover, as previously suggested, the NAc is more of a place where information integrated instead of a place that is purely responsive to reward or cost. 

### Updating the Cost Value with Learning

As the aforementioned dopaminergic neurons could code for the motivational value of a task, one might expect that they could also encode cost by lowering the neuron firing rate and, therefore, the motivation. However, the emerging evidence seems to suggest that the dopaminergic system encodes expected the reward value more than the anticipated cost [92]. The NAc dopamine release in rats is not affected by the manipulation of the response cost unless the response cost was reduced, and the effect only persisted in the first trial [49]. This is consistent with data from Hollon et al. [93], who found that the effort-related change in dopamine release was smaller than the anticipation of future reward and was more prominent in the initial few trials. Moreover, the proportion of neurons that are responsive to effort is not as significant. In the substantia nigra pars compacta (SNpc), only about 13% of dopamine neuron’ activity is discounted by effort, and about 47% of neurons only encode reward [94]. Depending on the neurons that the experimenter measures, different neurons may provide different responses. Despite their findings, the differential sensitivity to costs might simply be the distributional coding of dopaminergic neurons [38]: while some neurons are more ‘optimistic’, they appear to be less sensitive to the costs, and ‘pessimistic’ neurons appear to be more sensitive to costs.

Even though the dopaminergic neurons seem to be less sensitive to cost-related stimuli, they appear to be essential for learning of the cost of actions similar to the way they are essential for the learning of rewards [95]; however, this seems to be dependent on the modulation of the lateral habenula (LHb) and other brain structures [12,13]. Interest in the LHb has surged after Hikosaka et al. discovered that it might encode the negative prediction error and might indirectly modulate the dopaminergic neurons in the VTA and SNc [12]. Later findings suggested that the LHb neurons were excited by the conditioned stimulus, which predicted the absence of rewards or the presence of punishment and their response to unconditioned stimuli were modulated by the punishment prediction error [13]. Nevertheless, as the author has emphasised, their encoding of the punishment prediction error was not perfect, and they were more sensitive to the positive punishment prediction error than the negative punishment prediction error. 

The fact that LHb neurons are largely glutamatergic [96] raised the question about how the LHb neurons could inhibit the SNc and VTA dopaminergic neurons. It was later [97] found that the rostromedial tegmental nucleus (RMTg) in primates received excitation from the LHb and inhibited the dopaminergic neurons in the VTA and SNc. Moreover, a recent finding showed that different brain regions synapsed with RMTg could contribute differentially to punishment learning and aversive valence encoding [98]. The prelimbic cortex, brainstem parabrachial nucleus, and LHb signalled the aversive cues, outcomes, and punishment prediction error to RMTg, respectively, and modulated the dopaminergic neurons in a triply dissociable manner [98]. Therefore, it seems that the inhibition of dopaminergic neurons is dependent on these important brain structures to allow the learning of the cost of an action or a stimulus.

How the dopaminergic neurons instruct learning is another question that remains to be elucidated, but there have been some attempts to try to answer this question. The opponent actor learning hypothesis states that the direct pathway and indirect pathway in the basal ganglia represent the positive and negative outcomes of an action [99]. The direct pathway promotes the execution of an action, and the indirect pathway inhibits it, so the balance between the two pathways determines whether the action will be conducted or not. Moller and Bogacz [95] built a learning model of how dopamine could modulate the synapses between cortical neurons and their corresponding pathway in basal ganglia on the basis of this hypothesis. In their model, when the mice pressed a lever and received no reward, no dopamine was released, and the synapse between the indirect pathway and its associated cortical neurons was strengthened such that the same action was less likely to be carried out again. If the mice received the rewards later, the positive prediction error caused dopamine release at the basal ganglia, which reduced the weight of indirect pathway and boosted the weight of direct pathway to enhance the action. This model fits well with the experimental data that they obtained. 

As midbrain dopaminergic neurons project to a variety of brain regions, it is likely that the prediction error signalled by them could direct learning in areas where costs and benefits are encoded such that a value could be updated with learning. For example, an fMRI study has shown that the human dorsomedial prefrontal cortex (dmPFC) and SN/VTA encode the effort-related prediction error, with provides strong evidence for the prediction error signal being broadcasted from the SN/VTA to the dmPFC [100]. The electrophysiological evidence from mice shows that aversive stimuli selectively increase the AMPA/NMDA receptor ratio in the dopaminergic neurons projecting to the medial PFC, while rewarding stimuli that modify the dopaminergic neurons projecting to the NA medial shell [101]. It has been reported that some mesolimbic dopamine neurons do seem to encode the aversive stimuli and terminate at the ventral NAc medial shell where the reward-predicting excitation is excluded [102]. It should be noted that unlike the neurons with a firing rate discounted by the cost, these neurons show phasic firing to aversive stimuli. Although this type of dopaminergic neuron is not novel [103,104], it was believed that the projection of these neurons was mainly towards the medial PFC rather thantowards the NAc [105]. What is the molecular mechanism behind such learning? What are the functions of the dopaminergic neurons responding to aversive stimuli? What does it mean for animal decision making? These new findings pose more questions for future research to answer.

## 4. What Is the Optimal Strategy for the Task?

When the cost of a particular action has been considered, we can see a large degree of overlap between the area encoding benefits and costs. This is perhaps unsurprising because it would have some computational benefit, but it still brings out a new question: are these brain areas encoding the costs and benefits separately or are they encoding the net utility of task instead? It is surprising to see how many researchers have claimed that the brain regions that they study encode the net utility of a task or an action. There are so many studies similar to this that it makes one wonder whether net utility can really be encoded by so many brain regions. Perhaps it is not that surprising that we fail to find a one-to-one correspondence of function and structure [92]. Decision making is a very complicated process, and it involves variations in the number of attributes, the uncertainty of choice, the time of which the outcome is received, etc. It therefore should not be too surprising to see that according to the task that the experimenters examine, the brain region that appears to encode the net utility value of the task is different. Moreover, even when a choice is made, an agent needs to decide the actions that are appropriate for the task, which brings action selection into the picture.

### 4.1. Combined Value Calculation

Although the ACC has traditionally been associated with effort and cost, it also encodes reward [24]. Nevertheless, the picture is much more complicated. It does appear that apart from the neurons encoding costs and benefits, a subset of neurons in the ACC encodes uncertainty of either reward or punishment [79]. This shows that the ACC neurons are valence-specific, and the probability is represented separately. The average signal of all of these neurons could resemble a motivational salience signal for within-state decision making. Moreover, some researchers have suggested that what the ACC neurons encode is not reward or punishment but the prediction error instead [106], and the prediction error in turn affects the decisions of an agent.

However, in everyday life, decision making is not similar to those decisions that are investigated in the lab, in which all attributes are made available to an agent simultaneously. Foraging, for example, involves the decision that needs to be made across a longer time range, and the history during this process (e.g., the signs of predator or prey) would dynamically influence the choice (e.g., whether to go or to wait?). A decision made across time is called a state-change decision, as the attributes are constantly changing with time. The ACC neurons appear to be very important in state-change foraging tasks [24,79,107]. The evidence for this comes from ACC neuronal recording, which show that these neurons encode the task value of the chosen decision until the optimal option was discovered [24]. Therefore, these neurons can adjust their activities accordingly with the changing information in the external environment. These neurons also appear to be dynamically tracking the gains and losses during a foraging task [107], which is important in deciding whether to leave or to stay during foraging. With the ability to encode probability and to dynamically encode value, the ACC could integrate outcomes across a longer timescale and could help decision-making processes in a volatile environment [79]. In fact, the ACC may only be necessary in decision making when uncertainty is high, as the decision making of ACC-lesioned monkeys was not impaired when the weighting and integration of feedback over many timescales were less relevant [108,109]. However, if the probability is uncertain, then ACC could also guide action selection to seek more information and resolve uncertainties [110], which could be one of the functions of the connection to the associated motor cortex.

As mentioned in the previous section, the role of the OFC is to represent the task states and to allow access to the stored value when it is not obvious from the external stimuli [111]. This information could be conveyed to the ACC, where the probability could be used to weigh the attributes of decision. While some part of the OFC belong to vmPFC, the functional role of the vmPFC is not the same as that of the OFC. Instead, beyond the role of value-based decision making, the vmPFC could also regulate negative emotions and modulate social cognition [112]. It should be noted that there are also some researchers who do not make a great distinction between the OFC and the vmPFC [111]. With the close association between the ventral striatum and the amygdala, the vmPFC is central in value representation and expectation. Humans with vmPFC lesions appeared to have attenuated ventral striatum activity during reward anticipation [113], showing its role in value expectation. 

Interestingly, vmPFC has been associated with the somatic marker hypothesis proposed by Damasio, who claimed that emotions were used to guide decision making and that the vmPFC linked the action with its associated emotional outcome [109]. It is possible that the vmPFC could influence emotion given its dense connection to the amygdala, insula, dorsal ACC [114], and periaqueductal grey [115], and patients with vmPFC damage appeared to perform worse in gambling decision making tasks, as the somatic emotional cues were no longer available [109]. The somatic marker hypothesis has been challenged since then [116], and it now appears that although emotion does affect the decision making, the extent of this effect is dependent on an individual’s ability to detect their internal bodily state (i.e., interoception) [117]. 

Various studies show how different brain areas are involved in the decision-making process, but few discuss the molecular mechanism behind it. Recently, Hunt [14] suggested the frontal microcircuits in the prefrontal cortex and the anterior cingulate cortex (PFC/ACC) may support temporary extended cognitive decision. The microcircuits in the PFC/ACC are recurrent in nature and can sustain their activities over time through NMDA receptor-mediated effects that have a longer decay time constant [14]. The recurrent circuits receive inputs not only from other networks, but also from their own structure from a previous stage. This helps the agent to make a temporally extended decision in a natural environment where the evidence may accumulate over time. A value-based decision may arise from the mutual inhibition of different pools of neurons that represent different alternatives [14]. The pooled inhibition across the network allows the optimal decision to inhibit all of the other pools. Another property of these circuits is that they are widely distributed in many brain regions, which collectively form an interconnected system [15]. Value representation has an emergent property that is determined by pattern of connection instead of a scalar value. Therefore, different brain areas encode different aspects of the task, forming a parallel processing stream, and they complement each other to give rise to the ultimate value. This model is supported by experimental data from human studies using magnetoencephalography and animal models [15]. 

Dopamine in NAc has been suggested to play a key role in overcoming response costs and in exerting high effort to acquire reward [16]. This is possibly because it has dense connections to a variety of areas that influence value representation (e.g., VTA, frontal cortex), but it is also closely associated with areas related to memories and internal bodily states (e.g., hippocampus, amygdala) [105]. Moreover, the dopaminergic neurons projecting to NAc are closely associated with the lateral hypothalamus, where the internal deprivation state is counted [48]. The expressions of anorexigenic and orexigenic peptide receptors are also found in the VTA and SNpc, including insulin, orexin, gherkin, and leptin [118]. These factors act as indicator for the current body state and modulate behaviour accordingly. For example, it might be worth taking the risk to obtain food when animals are severely starved. Therefore, the NAc is a place where choice is weighted by internal bodily states and is energised by motivational value [16]. 

### 4.2. Action Selection

When the value has been calculated, and the choice has been made, the next step is to determine the appropriate action for performing the task. In simple operant conditioning, the action is reinforced by a reward, and action selection is less of an issue in model-free decision. However, in a model-based decision, choosing the right response from different alternatives becomes more important. This action selection is conducted by the basal ganglia (BG) [119]. The BG has been regarded as a place where actions are selected and where nonsense actions are filtered off. It receives input from the primary motor cortex which sends out the action programs to the BG. After the internal computation within the BG, the selected action output is sent back to the primary motor cortex via the thalamus such that the actions could be executed [120]. The opponent actor learning hypothesis states that there are two pathways within the BG: one is a direct pathway and the other is an indirect pathway. The direct pathway promotes the execution of the action, while the indirect pathway inhibits it [119,121]. The balance between the two pathways determines the ultimate output of the BG. The evidence for a such model comes from pathological conditions. Parkinson’s patients have dopamine depletion in the brain, and one of the major sites of impact is the substantia nigra, which sends the projection to the dorsal striatum and promotes the direct pathway while inhibiting the indirect pathway. The common symptoms seen in Parkinson’s patients are bradykinesia, akinesia, resting tremor, and muscle stiffness [122]. The inability to move voluntarily is caused by a striatal depletion of dopamine. On the other side, the enhancement of the direct pathway is seen in Huntington’s disease, which is a pathological loss of neurons in dorsal striatum. One of the key features of these patients is the execution of huge, involuntary movements [123]. This is due to the BG being unable to filter out the nonsense action programs sent by the primary motor cortex and being unable to inhibit their execution [123]. Nevertheless, this model has recently been reappraised, and the evidence shows that the division of the two pathways might not be that clear-cut after all [120].

## 5. Conclusions

How animals make decisions has always fascinated people, not only because of the use of this process in economy and business, but also for its wide implication in memory, learning, and central computation. This study could be useful for understanding the pathological conditions of human behaviours, such as impulsive behaviours following brain damage. Every decision comprises of many sub-components such as value prediction, choice selection, action selection, and learning. This means that multiple areas of the brain could be involved in every simple choice that we make, indicating the level of complexity in the decision-making computation. While forebrain structures have been implicated in task value representation and action selection in model-based decision making, the midbrain dopaminergic system counts for the internal bodily state and is involved in model-free decisions, although the division is less clear-cut. There are still a lot of questions left open for future investigation, and we are just revealing a part of the mystery of decision making.

## Figures and Tables

**Figure 1 brainsci-11-01096-f001:**
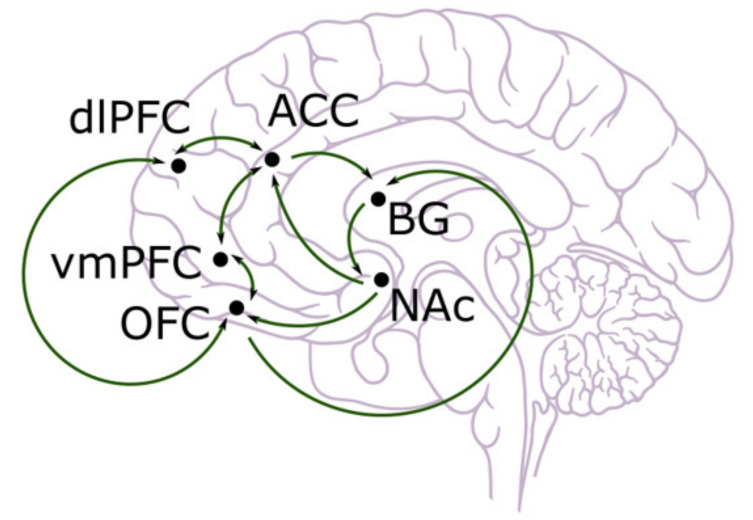
Anatomical illustration of dorsolateral prefrontal cortex (dlPFC), anterior cingulate cortex (ACC), ventromedial prefrontal cortex (vmPFC), orbitofrontal cortex (OFC), basal ganglia (BG), and nucleus accumbens (NAc) [55].

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
