# Peer review of "Neural Representation of Costs and Rewards in Decision Making"

_brainsci, 2021, doi:10.3390/brainsci11081096_

Round 1

Reviewer 1 Report

The manuscript provided a very impressive review on cost-benefit decision making. I have only a few minor comments.

  1. The author spent a lot of text on ramping vs. phasic DA signaling. However, one important work from the Uchida group is uncited (Kim 2020 Science), which directly contradicts Berke's motivation theory. It would be interesting to have a proper discussion on this.
  2. In the "cost" section, the author focuses on the OFC, while did not provide discussions on inhibitory inputs from LHb and RMTg onto DA neurons that drive punishment learning. This has been largely shown by Hikosaka group in monkeys (2006, 2007), Jhou group in rodents (2009, 2019a-c), and Malinow group, as well as other inputs that drive aversion (Lammel 2012).

Author Response

  1. The author spent a lot of text on ramping vs. phasic DA signaling. However, one important work from the Uchida group is uncited (Kim 2020 Science), which directly contradicts Berke's motivation theory. It would be interesting to have a proper discussion on this.   

Response: I'll have my article added with this. Thank you.

2. In the "cost" section, the author focuses on the OFC, while did not provide discussions on inhibitory inputs from LHb and RMTg onto DA neurons that drive punishment learning. This has been largely shown by Hikosaka group in monkeys (2006, 2007), Jhou group in rodents (2009, 2019a-c), and Malinow group, as well as other inputs that drive aversion (Lammel 2012).

Response: I'll discuss this in the article as well. Thnaks again for the comments.

Reviewer 2 Report

In this review, Chen emphasizes the complexity involved in decision making, action selection, reward and cost coding and finally, discusses how options are compared for the choice of the “optimal” decision. The paper reviews multiple processes involved in making such a decision, such as the system use of rewards, costs and motivation, this within types of decisions such as model free and model based, and incorporates the neural system involved, such as the dopaminergic system, the anterior cingulate cortex (ACC), the orbitofrontal cortex (OFC), the nucleus accumbens (NAc) etc. This is indeed a complicated task to achieve in one review paper which is bravely tackled by the author. It's worth noting that on the good side-the Autor do not attempt to answer all questions asked and the complexity of processes is well clarified. However, there several key issues that need to be addressed regarding the review:

  1. All through the text there are points which are discussed as known facts but are not referenced, or are un-based speculations which are not framed that way. For example: lines 60-61 – referring to the NAc role in weighting the internal states before action selection.

Lines 68-70 which refers to the “well established”  role of the dopaminergic system in reward encoding and pleasure.

Lines 77-81 – The whole framing of model free reinforcement learning , and the importance of dopamine in it.

Lines 90-93 – describing the dispute, and what is now “generally believed”.

These are just examples but the whole review is filled with them not allowing the reader to comfortably follow the line of thought, knowing that everything is well based and established.

  1. The description of model free and model based learning is not very clear with conflicting notions at different parts of the review and even seems inaccurate at times. For example the iclusion of both pavlovian and instrumental conditioning as being based solely on model free learning is very simplistic and is true in some instances, but not in others. As large portions of this review are based on this distinction perhaps a clearer view is needed.

  1. At several points the paper suggests a role of a system but does not elaborate enough on the subject so that the reader can fully understand the claim - e.g –the claim that both prediction error and motivation signals are encoded by phasic firing while distinction is done by striatal cholinergic interneurons switching dopaminergic synapses to "learning mode". Another example is the last topic of action selection which is the final step in the path the author sketches for the readers, but the transition in the text from the decision to the action is only very briefly discussed (p.18).

Minor point:

There are quite a few English errors, and editing problems. Since the paper is very elaborative and delivers complicated ideas – a thorough checkup and editing is needed.

Examples: rows 24-26, 63-64, 128 (units?) , 152 (reff-year) , 176-178, and more.

Author Response

  1. All through the text there are points which are discussed as known facts but are not referenced, or are un-based speculations which are not framed that way. For example: lines 60-61 – referring to the NAc role in weighting the internal states before action selection.

Response: thank you so much for the suggestion. I haven't got lots of help on the format of an academic paper, so I really appreciate that you've pointed this out. I'll have them fixed.

2. The description of model free and model based learning is not very clear with conflicting notions at different parts of the review and even seems inaccurate at times. For example the iclusion of both pavlovian and instrumental conditioning as being based solely on model free learning is very simplistic and is true in some instances, but not in others. As large portions of this review are based on this distinction perhaps a clearer view is needed.

Response: I'll fix this confusion and thanks for pointing that out.

3. At several points the paper suggests a role of a system but does not elaborate enough on the subject so that the reader can fully understand the claim - e.g –the claim that both prediction error and motivation signals are encoded by phasic firing while distinction is done by striatal cholinergic interneurons switching dopaminergic synapses to "learning mode". Another example is the last topic of action selection which is the final step in the path the author sketches for the readers, but the transition in the text from the decision to the action is only very briefly discussed (p.18).

Response: I'll try my best to make it clear!

Minor point:

There are quite a few English errors, and editing problems. Since the paper is very elaborative and delivers complicated ideas – a thorough checkup and editing is needed.

Examples: rows 24-26, 63-64, 128 (units?) , 152 (reff-year) , 176-178, and more.

Response: I'll fix this as well and thanks again.